# Evidential Stochastic Differential Equations for Time-Aware Sequential Recommendation

**Krishna Prasad Neupane, Ervine Zheng, Qi Yu**[*]
Rochester Institute of Technology
{kpn3569,mxz5733,qi.yu}@rit.edu

## Abstract

Sequential recommender systems are designed to capture users' evolving interests over time. Existing methods typically assume a uniform time interval among consecutive user interactions and may not capture users' continuously evolving behavior in the short and long term. In reality, the actual time intervals of user interactions vary dramatically. Consequently, as the time interval between interactions increases, so does the uncertainty in user behavior. Intuitively, it is beneficial to establish a correlation between the interaction time interval and the model uncertainty to provide effective recommendations. To this end, we formulate a novel Evidential Neural Stochastic Differential Equation (`E-NSDE`) to seamlessly integrate NSDE and evidential learning for effective time-aware sequential recommendations. The NSDE enables the model to learn users' fine-grained time-evolving behavior by capturing continuous user representation while evidential learning quantifies both aleatoric and epistemic uncertainties considering interaction time interval to provide model confidence during prediction. Furthermore, we derive a mathematical relationship between the interaction time interval and model uncertainty to guide the learning process. Experiments on real-world data demonstrate the effectiveness of the proposed method compared to the SOTA methods.

## 1 Introduction

Recommender systems have been used in various domains, such as e-commerce, entertainment, education, health care, social media, and many more [32, 11, 2, 44]. In these domains, users' interests and behaviors dynamically evolve over time. Therefore, capturing users' evolving behavior plays an essential role in effective recommendations. Various sequential recommendation models [35, 38, 40, 43] have been proposed accordingly. They mainly leverage the user's historical sequential interactions and aim to predict the next item that a user likes to interact with. These methods usually assume a uniform time interval of user interactions. However, in reality, the actual time interval between two consecutive interactions may vary dramatically, and a large time interval may be accompanied by a change in users' preferences.

The unrealistic assumption of a uniform interaction interval could significantly impact the model's capability to capture users' continuously evolving behavior and subsequently hurt the recommendation performance. Table 1 provides an illustrative example of the issues as outlined above. In this example, we follow the standard sequential recommendation similar to [19, 35] by predicting the next item considering

Table 1: Impact of interaction interval

| Interaction | Interval (Seconds) | Ranking ↓ | |
|---|---|---|---|
| | | **BERT4Rec** | **E-NSDE** |
| $6 \rightarrow 7$ | 44 | 4 | **4** |
| $13 \rightarrow 14$ | 623,591 | 24 | **16** |
| $116 \rightarrow 117$ | 62 | 6 | **3** |
| $150 \rightarrow 151$ | 896,291 | 56 | **18** |

---

[*]Corresponding author

38th Conference on Neural Information Processing Systems (NeurIPS 2024).

Table 2: Epistemic uncertainty and recommended movie genres for a random user considering sequential interaction intervals

| Model | Interval (Seconds) | Epistemic uncertainty | Ground-truth genre | Recommended genre |
|---|---|---|---|---|
| GRU-ODE | 44 | N/A | ['Comedy', 'Romance'] | ['Comedy', 'Drama'] |
| | 623,591 | N/A | ['Drama', 'Romance', 'War'] | ['Comedy', 'Romance'] |
| | 62 | N/A | ['Action', 'Adventure', 'Thriller'] | ['Action', 'Romance', 'Drama'] |
| | 896,291 | N/A | ['Horror', 'Thriller'] | ['Drama', 'Romance'] |
| **E-NSDE** | 44 | 0.4016 | ['Comedy', 'Romance'] | ['Comedy', 'Romance', **'Thriller'**] |
| | 623,591 | 0.6725 | ['Drama', 'Romance', 'War'] | ['Drama', **'Horror'**, **'Sci_Fi'**] |
| | 62 | 0.4463 | ['Action', 'Adventure', 'Thriller'] | ['Action', **'Mystery'**] |
| | 896,291 | 0.7104 | ['Horror', 'Thriller'] | ['Thriller', **'Crime'**, 'Adventure'] |

other 100 negative items from the item pool. We select a random user from the Movielens-100K dataset and provide recommended ranking results together with the interaction time intervals in the sequence. The recommendation performance for a sequential model BERT4Rec [35] deteriorates significantly when the gap between the consecutive interactions becomes large. For example, when the next interaction occurs soon after the first one, BERT4Rec usually ranks the ground-truth item in a top-10 list. However, for a much larger interval, the ground-truth item drops out of the top-20 or even the top-50 recommendation list.

Neural Ordinary Differential Equations (NODE) have been recently introduced that map the existing discrete neural networks to a continuous model [9, 36, 16], which can naturally capture users' continuously evolving preferences. However, one key limitation of these NODE models is the lack of uncertainty quantification capability, which is essential to understand user behavior when recommending the next item, especially when learning from extremely sparse user interactions given a large item space. Intuitively, the uncertainty of user behavior should increase along with the length of the interval since the last active interaction. For example, if the user has no activity for a long time, the user's prior interest tends to decrease. As a result, we become more uncertain about the user's preference. At the same time, a high uncertainty also presents a unique opportunity to effectively explore the item space by providing a diverse recommendation list to the users so that they may be attracted to a new category of items to keep them in the system.

Uncertainty can serve as a useful guidance to the recommender system to adapt to users' changing preferences and explore new items effectively. Table 2 further highlights the limitation of an existing ODE-based sequential recommendation method GRU-ODE [16], where recommended genres come from frequently watched 'Drama' and 'Romance' movies in the past. This issue becomes even worse when the interaction gap becomes large (*i.e.,* the fourth row of the GRU-ODE model), and the recommended genres significantly deviate from the ground truth. The recommendation list again concentrates on the same 'Drama' and 'Romance' genres, which are misaligned with the user's long-term interests. It is evident that conducting a more effective exploration is crucial to uncover the long-term and varied interests of users, aiming to optimize potential future benefits.

To address this critical gap as outlined above, we propose uncertainty augmented Neural Stochastic Differential Equations (NSDE) and integrate it into a novel sequential recommendation model. NSDE is a variant of NODE that adds Brownian

Table 3: Comparison of related recommendation models

| Key functions | Sequential | NODE | NSDE | E-NSDE |
|---|---|---|---|---|
| Varied interval | ✗ | ✓ | ✓ | ✓ |
| Uncertainty in preference | ✗ | ✗ | ✓ | ✓ |
| Uncertainty in interaction | ✗ | ✗ | ✗ | ✓ |

motion terms to incorporate stochasticity via a diffusion function. NSDE has been successfully applied in other fields, such as computer vision [20, 41]. It provides an effective means to capture users' continuously changing behavior, and also model the noise via stochasticity in user and item representations. However, recommender systems involve uncertainty from multiple sources: uncertainty caused by evolving user preferences and uncertainty arising from user-item interactions. While the standard NSDE can naturally capture the former, it is not designed to cover the latter. To this end, we further incorporate evidential deep learning (EDL) [33, 1] to gather evidence from user-item interactions and systematically capture uncertainty from multiple sources. Table 3 summarizes the key differences between the proposed model and existing relevant models.

The proposed E-NSDE seamlessly integrates an NSDE module and an EDL module, where the former is responsible for learning user and item representations over time and the latter utilizes these rich representations to identify important and diverse items that the model needs to learn to capture users' actual behavior with the help of uncertainty-aware exploration. Table 2 shows that E-NSDE places a pronounced emphasis on a wider array of genres, and a substantial portion of these manage

to capture sustained interest over the long term. In each sequence, E-NSDE attempts to provide diverse genres as shown in the first row ('Thriller') and second row ('Sci_Fi') of the E-NSDE result. When there is a long time interval, the model can leverage this opportunity guided by its predicted epistemic uncertainty and recommend a novel genre ('Crime') to explore the diverse items that can help to capture the user's long-term interest. The ability to quantify time-aware uncertainty also allows E-NSDE to outperform SOTA sequential recommendation models as shown in Table 1.

As a key innovation, we connect NSDE-based user-item representation learning with EDL using a monotonic network to provide an evidence-guided recommendation to capture time-sensitive rating prediction augmented with uncertainty estimation. Intuitively, the model's uncertainty of a user's interest should increase with respect to the time interval since the last observed user-item interaction. To model time-sensitive uncertainty, we integrate EDL using an uncertainty-aware regression model to infer evidential distributions that allow us to quantify both aleatoric and epistemic uncertainties. Similarly, the monotonic network captures the underlying constraint of monotonicity between the time interval and the predicted uncertainty: a longer time interval leads to a higher uncertainty. With accurately estimated uncertainty, the proposed framework effectively explores user preference from a large item space. The main contribution of this paper is fourfold:

- a novel recommendation model that integrates neural stochastic differential equations with evidential learning for time-aware uncertainty quantification for effective sequential recommendations,
- leveraging interaction and time-guided evidential uncertainty to maximize information gain through exploration of a large item pool,
- a monotonic network to ensure a positive correlation between interaction gap and uncertainty,
- an end-to-end integrated training process with seamless integration of NSDE and EDL modules.

To assess the feasibility of our method in comparison with existing state-of-the-art methods, we perform a wide range of experiments on publicly available real-world datasets. We also conduct ablation and case studies to analyze the effectiveness and interpretability of our method.

## 2 Related Work

In this section, we provide existing works most relevant to our proposed approach. Some additional related works are discussed in the Appendix.

**Sequential recommendation models.** Sequential models utilize users' historical interactions to capture users' preferences over time. Tang et al. utilized a CNN architecture to capture union level and point level contributions [37]. Kang et al. leveraged transformer-based user representation to better capture their interest [19]. Sun et al. utilized a bidirectional encoder for sequential recommendation [35]. Similarly, Zhou et al. [45] have leveraged auxiliary self-supervised objectives to learn correlation among attributes, items, sub-sequences, and sequences by utilizing mutual information maximization. Recently, contrastive learning has been used in sequential recommendation [40, 43] to learn high-quality user representation leveraging different forms of data augmentation strategies. Recently, SAR [3] has leveraged an actor-critic network, where the action is generated as adaptive sequence length to better represent the user's sequential pattern. Similarly, ResAct [42] utilizes residual actor-network to reconstruct policy that is close but better than online policy more efficiently in sequential recommendation. However, most existing sequential models are inadequate to capture long-term users' preferences, which is a critical gap that the proposed work aims to address.

**NODE based recommender systems.** Neural ODE solvers have recently been introduced into recommender systems [9, 36]. For example, the learnable time ODE-based collaborative filtering [9] redesigns linear graph convolution networks on top of the NODE that learns the optimal architecture and smooth ODE solutions for effective collaborative filtering. Similarly, [36] utilizes meta-learning enhanced neural ODE for citywide next POI Recommendation. It models city-invariant and city-specified information separately to achieve accurate citywide next POI recommendation. As discussed earlier, standard NODE models do not explicitly capture uncertainty, which is critical for a recommendation model to explore a large item space to capture user's long-term preference.

## 3 Preliminaries

**Problem formulation.** The input to a recommendation model include a user set ($U$) and item set ($I$), respectively. We represent a user as $u_t \in U$ and item as $i_t \in I$ at time $t$. In a sequential recommendation setting, users' interactions are organized in the chronological order and we use

$(i_0, i_1, ..., i_{t-1}, i_t)^\top$ to represent interaction sequences for user $u$ at time $t$. We perform recommendation and uncertainty quantification for each user using a function:

$$f_\Theta(u_t, i_t) = \left\{\gamma_{(u_t,i_t)}, \nu_{(u_t,i_t)}, \alpha_{(u_t,i_t)}, \beta_{(u_t,i_t)}\right\}$$

where $\gamma_{(u_t,i_t)}$ is the recommendation score for item $i$ assigned by user $u$, $\nu_{(u_t,i_t)}$ and $\alpha_{(u_t,i_t)}$ are the model evidence, and $\beta_{(u_t,i_t)}$ is a total uncertainty arising from the user-item interaction at time $t$.

**Evidential learning.** Evidential learning is an evidence acquisition process where every training sample adds support to learn a higher order evidential distribution [33, 1]. Given that the target $y_n$ is drawn i.i.d. from a Gaussian distribution with unknown mean and variance $(\mu, \sigma^2)$ the model evidence can be introduced by further placing a prior distribution on $(\mu, \sigma^2)$. Placing a Gaussian prior on the unknown mean and the Inverse-Gamma prior on the unknown variance, the posterior of $(\mu, \sigma^2)$ is the Normal-Inverse-Gamma (NIG) distribution. The Gaussian and the Inverse-Gamma priors are chosen to ensure conjugacy:

$$p(y_n|\mu, \sigma^2) = \mathcal{N}(\mu, \sigma^2), \ p(\mu|\gamma, \sigma^2\nu^{-1}) = \mathcal{N}(\gamma, \sigma^2\nu^{-1}), \ p(\sigma^2|\alpha, \beta) = \text{Inv-Gamma}(\alpha, \beta)$$

where $\text{Inv-Gamma}(z|\alpha, \beta) = \frac{\beta^\alpha}{\Gamma(\alpha)}\left(\frac{1}{z}\right)^{\alpha+1}\exp(-\frac{\beta}{z})$ with $\Gamma(\cdot)$ being a gamma function; $\mathbf{m} = (\gamma, \nu, \alpha, \beta)$ are parameters of the corresponding prior distributions. The posterior of $(\mu, \sigma^2)$ follows a Normal Inverse-Gamma (NIG):

$$p(\mu, \sigma^2|\mathbf{m}) = \frac{\beta^\alpha\sqrt{\nu}}{\Gamma(\alpha)\sqrt{2\pi\sigma^2}}\left(\frac{1}{\sigma^2}\right)^{\alpha+1}\exp\left\{-\frac{2\beta + \nu(\gamma-\nu)^2}{2\sigma^2}\right\}$$

Given a NIG posterior, one can derive the mean ($\mathbb{E}[\mu]$), aleatoric ($\mathbb{E}[\sigma^2]$) and epistemic ($\text{Var}[\mu]$) uncertainty as:

$$\mathbb{E}[\mu] = \gamma, \ \mathbb{E}[\sigma^2] = \frac{\beta}{\alpha-1}, \ \text{Var}[\mu] = \frac{\beta}{\nu(\alpha-1)} \tag{1}$$

**Neural ordinary differential equations (NODE).** Ordinary Differential Equations (ODEs) are used to model continuous-time hidden dynamics in neural networks [7] that can be defined as:

$$dh_t = f_\psi(h_t, t)\mathrm{d}t, \quad h_0 \in \mathcal{R}^d \tag{2}$$

where $f(\cdot)$ is a neural network with parameter $\psi$ and $h_0$ is an initial value. Leveraging Eq (2) and integrating these dynamics forward, one can compute $\mathbf{h}(t_{i+1})$ from $\mathbf{h}(t_i)$ by solving the following Riemann integral problem:

$$\mathbf{h}(t_{i+1}) = \mathbf{h}(t_i) + \int_{t_i}^{t_{i+1}} f(\mathbf{h}(t_i), t; \psi)dt \tag{3}$$

**Neural stochastic differential equations (NSDE).** We could view Stochastic Differential Equations (SDE) as an ODE with infinitesimal noise added throughout time:

$$dh_t = f_\psi(h_t, t)dt + g_\omega(h_t, t)dB_t \tag{4}$$

where $f(.)$ and $g(.)$ are drift and diffusion functions respectively, $B_t$ is a Brownian motion. Similar to NODE, we are also able to compute the forward dynamics of NSDE i.e. $\mathbf{h}(T)$ from initial value $\mathbf{h}(t_0)$ integrating Eq (4) as:

$$h(T) = h(t_0) + \int_{t_0}^{T} f(h(t), t; \psi)dt + \int_{t_0}^{T} g(h(t), t; \omega)dB_t \tag{5}$$

## 4 Time-Aware Sequential Recommendations

**Overview.** We propose a novel time-aware sequential recommendation model as shown in Figure 1. The model leverages 1) NSDE to capture continuous time-evolving user dynamics and 2) an evidential module to capture uncertainty in user-item interactions and also to provide uncertainty-aware exploration that takes into consideration the interaction interval. The NSDE module takes the initial representations of users and items as inputs, and uses the interaction time gap to generate refined user and item representations. Subsequently, these improved user and item representations are fed into the EDL module. The rating network then generates a rating score, while the monotonic network produces evidential parameters that incorporate the interaction time gap, establishing a direct link to the model's predicted uncertainty. Our approach adheres to the conventional sequential training strategy and incorporates supervised signals derived from evidential learning. We delve into detailed discussions in the subsequent sections.

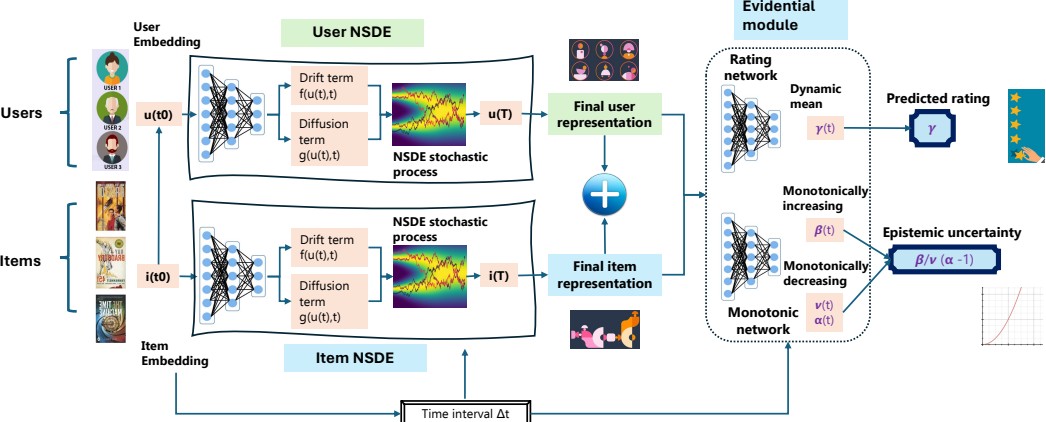

Figure 1: Overview of E-NSDE framework, which includes user and item NSDE modules to generate the final user and item representation and an EDL module to provide an uncertainty-aware prediction.

## 4.1 NSDE Based User and Item Representations

The NSDE includes two key components: drift and diffusion functions. The drift component captures the system's evolving nature, and the diffusion component captures its stochasticity. The proposed NSDE for recommender system advances contemporary sequential models from the following aspects: 1) existing methods require partitioning the time into uniform intervals to support model training and inference, while NSDE removes this requirement, providing additional flexibility; 2) the existing methods largely negate stochasticity in the system, but the NSDE incorporates it into the form of inherent noise. The above issues suggest a better fit of the NSDE into sequential recommendation and essentially support generating richer user and item representations. This is because SDE can capture users' continuously evolving preferences over time, while previous discrete sequential and deterministic ODE methods cannot. The fine-grained user representation (based on NSDE formulation) can be written following (5):

$$u(T) = u(t_0) + \int_{t_0}^{T} f(u(t), t; \psi) dt + \int_{t_0}^{T} g(u(t), t; \omega) dB_t \tag{6}$$

where $u(t_0)$ is the user's initial representation which aggregates a set of initially interacted items: $u(t_0) = \mathrm{agg}(i_1..i_k)$ and $T$ is the final time.

In the setting of recommender systems, the drift component naturally encodes the evolution of the user's preference, while the diffusion component captures inherent noise that occurs when the user interacts with the environment. In particular, we leverage the diffusion function with Brownian motion to capture the user's inherent noise over each interaction. This term is crucial in SDE to capture stochasticity in the system. By capturing stochasticity, we incorporate the impact of noisy user-item interactions. As it includes the Brownian motion for stochasticity, we relate this in recommender systems to incorporate noise considering the time interval of interaction. We first provide the standard definition and properties of the Brownian motion and then present its applicability in the recommendation setting.

**Definition 1** (Standard Brownian Motion). *A standard Brownian motion $B_t$ is a stochastic process that satisfies the following properties: a) $B_t - B_s$ is normally distributed with zero mean and $(t - s)$ variance: $\mathcal{N}(0, t - s)$ for all $t \geq s \geq 0$; b) For every pair of disjoint time intervals $[t_1, t_2]$ and $[t_3, t_4]$, with $t_1 < t_2 \leq t_3 \leq t_4$, the increments $B_{t_2} - B_{t_1}$ and $B_{t_4} - B_{t_3}$ are independent random variables.*

**Theorem 1.** *For user representation defined by the Brownian motion in Eq.(6), with two adjacent timestamps of interaction denoted as $t_2$ and $t_1$, a larger interaction time interval $(t_2 - t_1)$ guarantees a higher uncertainty of user representation.*

Given Theorem 1, the final term in Eq. (6) captures the user's time deviation and its impact in increasing large variance or noise in the system. Further, the first term captures the user's initial representation with some interactions, and the second drift component provides the user's evolving

interest. Considering all of these three components, the SDE solver captures the user's richer representation over time. Similarly, NSDE-based item representation processes can be formulated as:

$$i(T) = i(t_0) + \int_{t_0}^{T} f(i(t), t; \psi)dt + \int_{t_0}^{T} g(i(t), t; \omega)dB_t \tag{7}$$

where, $i(t_0)$ represents item's initial representation

## 4.2 Evidential Module

We leverage an evidential learning technique to provide an uncertainty-aware model prediction for effective recommendations. The evidential module consists of two key networks: Rating Network and Monotonic Network.

**Rating Network.** The rating network utilizes the fine-grained user $u_t$ and item $i_t$ representations from the NSDE and predicts the score $\gamma_{(u_t, i_t)}$ of the corresponding user item interactions. We adopt an evidential loss as the marginal likelihood while computing the predicted loss. This includes the negative log-likelihood ($\mathcal{L}^{NLL}[f_\Theta]$) to maximize the marginal likelihood and an evidential regularizer ($\mathcal{L}^{R}[f_\Theta]$) to impose a high penalty on the predicted error with low uncertainty (i.e., high confidence). We first formulate the negative log-likelihood, given by

$$\mathcal{L}^{NLL}[f_\Theta(u_t, i_t)] = -\log(p(r_{(u_t, i_t)}|\mathbf{m}(u_t, i_t)) \tag{8}$$

where, $\mathbf{m}(u_t, i_t) = (\gamma_{(u_t, i_t)}, \nu_{(u_t, i_t)}, \alpha_{(u_t, i_t)}, \beta_{(u_t, i_t)})$ are model parameters at time t, and $p(r_{(u_t, i_t)}|\mathbf{m}(u_t, i_t)) = St(r_{(u_t, i_t)}; \gamma_{(u_t, i_t)}, \frac{\beta_{(u_t, i_t)}(1+\nu_{(u_t, i_t)})}{\nu_{(u_t, i_t)}\alpha_{(u_t, i_t)}}, 2\alpha_{(u_t, i_t)})$ is a student t-distribution acquired after placing a NIG evidential prior on Gaussian likelihood function. We formalize our own evidence regularizer, which considers epistemic uncertainty to penalize confidently predicted errors. We multiply the predicted error with the inverse epistemic uncertainty that scales up the error, which encourages high inverse epistemic uncertainty when the predicted evidence is high (and vice-versa). Conversely, it will be less penalized if the prediction is close to the target score:

$$\mathcal{L}^{R}[f_\Theta(u_t, i_t)] = |r_{(u_t, i_t)} - \gamma_{(u_t, i_t)}| \cdot \left[ \frac{\nu_{(u_t, i_t)}(\alpha_{(u_t, i_t)} - 1)}{\beta_{(u_t, i_t)}} \right] \tag{9}$$

The regularized EDL loss for each sequential update is:

$$\mathcal{L}_{EDL}(u_t, i_t) = \mathcal{L}^{NLL}[f_\Theta(u_t, i_t)] + \lambda \mathcal{L}^{R}[f_\Theta(u_t, i_t)] \tag{10}$$

where $\lambda$ is a regularization coefficient.

**Monotonic Network.** We adopt the concept of a monotonic network [34] in the context of building the relationship between the interaction time gap and model uncertainty. Intuitively, the monotonic network is designed in such a way that the increase in input, i.e., time interval ($\Delta t$), increases the output, i.e., the variance of the predicted rating. The variance or epistemic uncertainty is computed as given by Eq (1). For this, the nominator term, i.e., total uncertainty $\beta_{(u_t, i_t)}$ should need to be increased with the increase in $\Delta t$, and the denominator terms, i.e., pseudo-observations $\nu$ and $\alpha$ should need to be decreased with the increase in $\Delta t$. We theoretically show this intuition in the following theoretical section and show a mathematical relation. We maintain this by performing exponential transformation of the network weights as: $\phi = \exp(\phi_{init})$, where $\phi_{init}$ is the network's initial weight. To ensure $\beta$ to monotonically increase, we assign all positive weights to the network layers. Similarly, to ensure $\nu$ and $\alpha$ to monotonically decrease, we assign negative weights to the last layer and positive weights to other layers of the network. We update the network utilizing the total loss similar to the rating network.

**Lemma 1** (Monotonic increase of total uncertainty $\beta$). *Let the total uncertainty of user-item interaction $\beta_{(u_t, i_t)}$ be the output of the evidential monotonic network with weights $\phi = \exp(\phi_{init})$. Given a time interval $\Delta t$, the output of the network is guaranteed to monotonically increase.*

**Lemma 2** (Monotonic decrease of pseudo-observations $\alpha$ and $\nu$). *Let $\Delta t$ be the increased time interval $\Delta t$, weights $W_L$ of the last layer be negative, and weights $W_{0,...,L-1}$ for other layers be positive, the output i.e., pseudo-observations $\alpha_{(u_t, i_t)}$, and $\nu_{(u_t, i_t)}$ of the evidential monotonic network decreases monotonically.*

**Theorem 2.** *(Increased time interval $\Delta t$ results in increased in epistemic uncertainty $Var[\mu]$). Given the monotonic network formulated by Lemmas 1 and 2, an increase of the input time interval ($\Delta t$) of the evidential monotonic network guarantees an increase of the output epistemic uncertainty $Var[\mu]$.*

*Proof.* The epistemic uncertainty equation from Eq (1):

$$\mathcal{U}_{(u_t,i_t)} = \text{Var}[\mu] = \frac{\beta_{(u_t,i_t)}}{\nu_{(u_t,i_t)}(\alpha_{(u_t,i_t)} - 1)}$$

Given the increase $\beta_{(u_t,i_t)}$ from Lemma 1 and decrease in $\alpha_{(u_t,i_t)}$ and $\nu_{(u_t,i_t)}$ from Lemma 2 the nominator of the epistemic uncertainty increases, and the denominator decreases. This proves that the increase in the time interval ($\Delta t$) increases the epistemic uncertainty var$[\mu]$ of the evidential monotonic network. We enforce the constraints on $(\beta_{(u_t,i_t)}, \alpha_{(u_t,i_t)}, \nu_{(u_t,i_t)})$ with a `soflplus` activation and adding 1 to $\alpha_{(u_t,i_t)}$ since $\alpha_{(u_t,i_t)} > 1$).  □

**Interpreting hyper-parameters.** Besides serving as the parameters of the evidential prior distributions, the hyper-parameters $(\nu_{(u_t,i_t)}, \alpha_{(u_t,i_t)}, \beta_{(u_t,i_t)})$ offer very intuitive meanings. First, both $\nu_{(u_t,i_t)}$ and $\alpha_{(u_t,i_t)}$ are essentially the 'pseudo' prior observations, and their posterior can be treated as the *evidence* to support a prediction. In the context of the recommendation, their relation with time interval is inverse, because the large time gap causes a decrease in the number of pseudo items, as mentioned in Lemma 2. Second, the $\beta_{(u_t,i_t)}$ hyperparameter combines total uncertainty from pseudo samples and observed data. Lemma 1 shows that an increase in time interval will result in an increase in the uncertainty (due to a smaller number of pseudo and interacted items to the user), and therefore, the model will be less confident in providing an accurate prediction.

**Weighted Bayesian personalized ranking (WBPR) loss.** To leverage the effective exploration for the long-term, we formulate weighted BPR loss which is computed from non-interacted (*i.e.,* negative) items, $j_t \in \mathcal{N}_t$ that are similar to the user's future interacted items. We first select similar negative items from the user non-interacted item pool and then leverage cosine similarity with future positive item embeddings. Further, the model provides uncertainty-aware predicted rating score for those negative items leveraging both rating and monotonic network output as:

$$\hat{r}_{(u_t,j_t)} = \gamma_{(u_t,j_t)} + \eta\mathcal{U}_{(u_t,j_t)} \tag{11}$$

where $j_t$ represents non-interacted items at time t and $\eta$ is scalar to control the influence of epistemic uncertainty. We then compute weight coefficients based on uncertainty-aware predicted scores with cosine similarity $\text{Sim}(\cdot)$ as:

$$w_{(i_t,j_t)} = \begin{cases} \max[\text{Sim}(\text{f\_emb, j\_emb})], \text{if } \hat{r}_{(u_t,j_t)} > \tau \\ \min[\text{Sim}(\text{f\_emb, j\_emb})], \text{otherwise} \end{cases}$$

where $f\_emb, j\_emb$ are future and negative item embedding, respectively. We then formulate weighted BPR loss utilizing a negative log-likelihood function as:

$$\mathcal{L}_{\text{WBPR}}(u_t, i_t) = \sum_{(u_t,i_t,j_t \in \mathcal{N}_t)} w_{(i_t,j_t)}\{-\ln[\sigma(\hat{r}_{(i_t,j_t)})]\} \tag{12}$$

where $\hat{r}_{(i_t,j_t)} = \hat{r}_{(u_t,i_t)} - \hat{r}_{(u_t,j_t)}$, $\sigma(\cdot)$ is the sigmoid.

**Remark.** The intuition behind this weighted BPR formulation is to learn effective exploration by providing higher weight to the non-interacted items which are quite similar to user future positive items so that model can learn quickly a diverse range of evolving behavior to benefit the future.

The overall loss of the end-to-end model training is obtained by combining the EDL and WBPR loss:

$$\mathcal{L}(u_t, i_t) = \mathcal{L}_{\text{EDL}}(u_t, i_t) + \zeta\mathcal{L}_{\text{BPR}}(u_t, i_t) \tag{13}$$

where $\zeta$ represents the balancing factor between EDL and WBPR loss. Training and inference details are provided in Appendix D.

# 5 Experiments

**Experimental setup.** Our experiment setting of sequential recommendation is based on next-item recommendation tasks, which was used in [35]. We first split users by 70% into train and 30% in

Table 4: Recommendation performance comparison

| Category | Model | MovieLens-100K | | Movielens-1M | | Netflix | | Amazon Book | |
|---|---|---|---|---|---|---|---|---|---|
| | | P@5 | nDCG@5 | P@5 | nDCG@5 | P@5 | nDCG@5 | P@5 | nDCG@5 |
| Dynamic MF | timeSVD++ | 0.3842±0.015 | 0.3420±0.013 | 0.3917±0.016 | 0.3508±0.013 | 0.3756±0.016 | 0.2951±0.013 | 0.3601±0.014 | 0.3128±0.012 |
| | CKF | 0.3916±0.017 | 0.3620±0.013 | 0.3928±0.016 | 0.3552±0.015 | 0.3600±0.017 | 0.2986±0.014 | 0.3823±0.016 | 0.3214±0.015 |
| Graph | NGCF | 0.3859±0.014 | 0.3662±0.012 | 0.3978±0.016 | 0.3587±0.018 | 0.3574±0.015 | 0.3167±0.017 | 0.3574±0.012 | 0.3321±0.011 |
| | LightGCN | 0.4103±0.014 | 0.3702±0.013 | 0.4028±0.017 | 0.3632±0.015 | 0.3617±0.013 | 0.3204±0.016 | 0.3678±0.013 | 0.3382±0.012 |
| Sequential | CASER | 0.4096±0.012 | 0.3663±0.015 | 0.4021±0.014 | 0.3626±0.016 | 0.3658±0.013 | 0.3189±0.012 | 0.3722±0.012 | 0.3414±0.012 |
| | SASRec | 0.4105±0.013 | 0.3740±0.011 | 0.4112±0.015 | 0.3708±0.017 | 0.3746±0.012 | 0.3257±0.014 | 0.3812±0.014 | 0.3445±0.012 |
| | BERT4Rec | 0.4149±0.014 | 0.3781±0.011 | 0.4163±0.012 | 0.3754±0.013 | 0.3793±0.011 | 0.3295±0.013 | 0.3846±0.013 | 0.3463±0.013 |
| | $S^3$-Rec | 0.4124±0.012 | 0.3755±0.014 | 0.4134±0.013 | 0.3715±0.014 | 0.3786±0.016 | 0.3274±0.013 | 0.3725±0.014 | 0.3350±0.012 |
| | CL4SRec | 0.4210±0.016 | 0.3821±0.017 | 0.4205±0.013 | 0.3781±0.015 | 0.3814±0.014 | 0.3318±0.012 | 0.3858±0.014 | 0.3313±0.011 |
| | SAR | 0.4034±0.012 | 0.3741±0.012 | 0.4023±0.014 | 0.3747±0.014 | 0.3711±0.012 | 0.3224±0.013 | 0.3658±0.012 | 0.3320±0.014 |
| | ResAct | 0.4366±0.014 | 0.3948±0.015 | 0.4286±0.014 | 0.3814±0.011 | 0.3867±0.012 | 0.3360±0.011 | 0.3884±0.013 | 0.3472±0.013 |
| ODE | LT-OCF | 0.4267±0.013 | 0.3785±0.015 | 0.4141±0.016 | 0.3673±0.014 | 0.3848±0.012 | 0.3313±0.013 | 0.3841±0.014 | 0.3416±0.012 |
| | GRU-ODE | 0.4398±0.014 | 0.3902±0.017 | 0.4275±0.013 | 0.3792±0.012 | 0.3994±0.013 | 0.3417±0.015 | 0.3856±0.014 | 0.3455±0.012 |
| **Proposed** | **E-NSDE** | **0.4711±0.015** | **0.4112±0.013** | **0.4551±0.011** | **0.3982±0.016** | **0.4194±0.013** | **0.3637±0.015** | **0.4021±0.014** | **0.3621±0.012** |

test. For each user, we leverage the fixed sequence length and hold out the next item of the behavior sequence as the target item. We follow the standard strategy in [35] for easy and fair evaluation. We leverage the actual time of interactions (in UNIX timestamp) to provide user preference evolution.

**Datasets and baselines.** We conduct extensive experiments on four real-world datasets that contain explicit ratings: *Movielens-100K*, *Movielens-1M*, *Netflix*, and *Amazon Book*.

- **Movielens-100K**[2]: This dataset contains 100,000 explicit ratings on a scale of (1-5) from 943 users on 1,682 movies. Each user at least rated 20 movies from September 19, 1997 through April 22, 1998.
- **Movielens-1M**[3]: This dataset includes 1M explicit feedback (i.e. ratings) made by 6,040 anonymous users on 3,900 distinct movies from 04/2000 to 02/2003.
- **Netflix** [4]: This dataset has around 100 million interactions, 480,000 users, and nearly 18,000 movies rated between 1998 to 2005. We pre-processed the dataset and selected 6,042 users with user-item interactions from 01/2002 to 12/2005.
- **Amazon Book** [39]: This data set contains 2,984,108 ratings applied to 91,599 books by 52,643 users with at least ten interactions in each user sequence

For comparisons, we include a comprehensive list of SOTA baselines from diverse groups, including *Dynamic models:* timeSVD++ [22] and CKF [14]; *Sequential models:* CASER [37], SASRec [19], BERT4Rec [35], $S^3$-Rec[45], CL4SRec [40], SAR [3], and ResAct [42]; *Graph-based models:* NGCF[39] and LightGCN[18]; *ODE-based models:* LT-OCF [9] and GRU-ODE [16].

**Evaluation metrics.** To evaluate the proposed and baseline recommendation models, we follow the sequential recommendation setup similar to [35]. We consider one ground truth item in each sequential recommendation. We use two standard metrics to measure the recommendation performance.

- **Precision@N (P@N)**: It is the fraction of the top-$N$ items recommended in each sequence to the user . We reported the average overall sequence precision value as the final precision. Further, due to only one ground truth in the target, the P@N is equivalent to Recall@N.
- **nDCG@N** : Normalized Discounted Cumulative Gain (nDCG) measures ranking quality, considering the relevant items within the top-$N$ of the ranking list in each recommendation.

For more details about datasets, and implementation please refer to the Appendix.

**Recommendation performance comparison.**  Table 4 summarizes the recommendation performance from all models for four real-world datasets. The proposed model benefits from both the SDE module, which continuously captures user evolving preferences, and the evidential module, which estimates prediction confidence, and thus achieves better results in all four datasets. The dynamic models achieve less ideal performance due to their focus on discrete-term user interest and inability to provide continuous user preference. Graph-based methods take advantage of recently interacted items and have shown better performance than traditional dynamic methods. However, they may not be good enough to capture user sequential interest. Further, sequential methods benefit from sequential learning and have promising results. However, they do not consider the time component in the recommendation and are less effective than the proposed method. ODE-based methods have

---

[2] https://grouplens.org/datasets/movielens/100k/
[3] https://grouplens.org/datasets/movielens/1M/

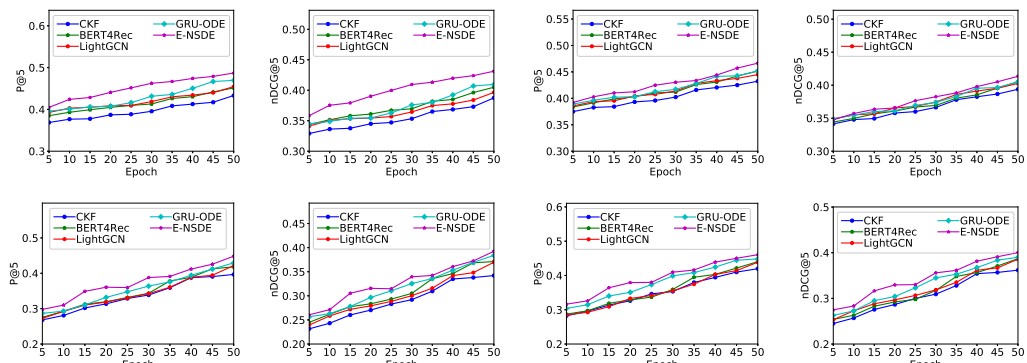

Figure 2: P@5 and nDCG@5 Performance: MovieLens-100K, MovieLens-1M, Netflix, and Amazon from top to down.

Table 5: (a) Diverse recommendations by E-NSDE; (b) Ablation of key components

| Model | Important Movies (Genre) | Future Movie's Genre |
|---|---|---|
| **GRU-ODE** | Dead Man Walking ('Drama') | 'Thriller' |
| | Richard III ('Drama') | 'Drama' |
| | Mad Love ('Romance') | 'Mystery' |
| **E-NSDE** | GoldenEye ('Thriller') | 'Crime' |
| | Taxi Driver ('Drama') | 'Sci_Fi' |
| | Twelve Monkeys ('Sci_Fi') | |

(a)

| NSDE | EDL | WBPR | Performance | |
|---|---|---|---|---|
| | | | P@5 | nDCG@5 |
| ✓ | ✗ | ✗ | 0.4065 | 0.3677 |
| ✓ | ✓ | ✗ | 0.4523 | 0.3962 |
| ✓ | ✗ | ✓ | 0.4120 | 0.3715 |
| ✓ | ✓ | ✓ | **0.4711** | **0.4112** |

(b)

shown a clear advantage due to their focus on capturing users' continuous behavior over time, but they cannot estimate model confidence on predictions and hence have lower performance value than the proposed method. We provided a detailed plot of precision and nDCG @5 in Figure 2 considering test users in each training epoch. Further, we report the precision and NDCG @10 and @20 as the increased top-$N$ metric (*i.e.,* $N = 10, 20$) for several baselines in the appendix. The results show the proposed model is consistently outperforming other baselines.

**Uncertainty vs. Interaction gap** We further investigate the impact of the user-item interaction gap and the corresponding uncertainty in providing important and diverse items. Table 5 (a) shows the proposed E-NSDE model provides diverse (*i.e.,* 'Thriller' and 'Sci_Fi') movies for the larger interaction time gap $\Delta t = 896, 291$ seconds that help to capture the future interest of the user. On the other hand, the existing GRU-ODE model recommends only popular genres like 'Drama' and fails to explore others adequately. We further provided the simulation results of 5 users considering increasing time intervals:

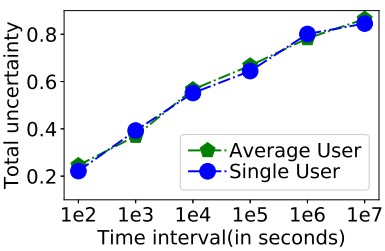

Figure 3: $\beta$ vs. $\Delta t$

$\{10^2, 10^3, 10^4, 10^5, 10^6, 10^7\}$ in seconds with random item interactions on those timestamps and the corresponding predicted uncertainty in Figure 3. The total uncertainty $\beta$ monotonically increases with the time intervals which shows that E-NSDE effectively captures user's uncertainty coming from longer interaction gaps.

**Effectiveness of exploration.** To demonstrate the effectiveness of the uncertainty-based exploration, we measure the diversity of the recommended items. We consider three commonly used diversity measures: Gini index ($G$), Coverage ($C$), and Novelty ($N$):

- **Gini index** can be calculated by leveraging the different genres as categories for each recommended top $K$ items in a sequence: $G = 1 - \sum_{c=1}^{C} P(c)^2$, where $C$ is the number of categories and $P(c), c \in [1, C]$ is the probability for each category.
- **Coverage** represents the number of categories that are included in the recommendation list. For the recommended top $K$ items, we compute item coverage as: $C = \#$ categories/total categories.
- **Novelty** is defined by the ratio of recommended items from new categories not interacted by the user. It is computed as : $N = \#$ new categories $/K$.

Table 6: Gini index, coverage, and novelty comparisons

| Models | Gini | | Coverage | | Novelty | |
|---|---|---|---|---|---|---|
| | MovieLens-1M | Netflix | MovieLens-1M | Netflix | MovieLens-1M | Netflix |
| BERT4Rec | 0.6845 | 0.6926 | 0.4103 | 0.3718 | 0.1434 | 0.1752 |
| GRU-ODE | 0.6632 | 0.6784 | 0.4142 | 0.3812 | 0.1482 | 0.1866 |
| **E-NSDE** | **0.7413** | **0.7811** | **0.4416** | **0.4026** | **0.2402** | **0.2236** |

(a) Embedding size      (b) Uncertainty-aware rating      (c) WBPR loss

Figure 4: Average nDCG@5 plot for different embedding sizes, and balancing factors

Table 6 shows that E-NSDE recommends more diverse items based on all three evaluation metrics as compared with the two competitive baselines thanks to its uncertainty-guided exploration strategy.

Table 7: Average P@5 and nDCG@5 for *E-NSDE* with different regularizer values

| Regularizer | MovieLens 1M | | Netflix | |
|---|---|---|---|---|
| ($\lambda$) | **P@5** | **nDCG@5** | **P@5** | **nDCG@5** |
| 0 | 0.4236 | 0.3786 | 0.4022 | 0.3574 |
| 0.0001 | 0.4412 | 0.3914 | 0.4134 | 0.3615 |
| **0.001** | **0.4551** | **0.3982** | **0.4194** | **0.3637** |
| 0.01 | 0.4518 | 0.3942 | 0.4096 | 0.3605 |
| 0.1 | 0.4224 | 0.3756 | 0.4064 | 0.3586 |
| 1 | 0.4116 | 0.3711 | 0.3923 | 0.3502 |

**Ablation study.** We conduct an ablation study and details are summarized below:

- *Impact of key components.* We evaluate the impact of each key component in the proposed E-NSDE. Table 5 (b) shows each component contributes to improved recommendation performance.
- *Evidential regularization parameter.* One of the key hyperparameters of the *E-NSDE* model is the regularizer constant ($\lambda$) for the evidential learning. We cross-validated this parameter with empirical results of the model for the different $\lambda$ values in two datasets as shown in Table 7. From the table, our model achieves the best performance in both datasets with $\lambda = 0.001$.
- *Embedding dimension.* We generate user and item embeddings using the embedding network. We perform a grid search for the embedding dimension ($d$) of the user and item representation in *E-NSDE* model as shown in Figure 4a. From the plot, it shows that E-NSDE has the best performance with $d = 64$.
- *Balancing factors.* We leverage grid search on uncertainty-aware ranking factor $\eta$, and WBPR loss balancing factor $\zeta$ on three datasets as shown in Figure 4b and Figure 4c, respectively. The figure shows a clear advantage with $\eta = 0.01$, which indicates that the uncertainty-aware exploration component takes an effective role in providing the best performance for our proposed E-NSED model. Similarly, for $\zeta$ balancing factor integrated overall loss has the best performance when it is equal to $0.001$.

# 6 Conclusion

We propose a novel evidential stochastic differential equations (E-NSDE) model for the time-aware sequential recommendations. E-NSDE seamlessly integrates an NSDE module and an EDL module to capture users' continuously evolving behavior and model predictive uncertainty at the same time. Our proposed model effectively leverages the interaction time gap and provides uncertainty-aware recommendations with diverse items to the user. Further, we theoretically derive mathematical relationships between the interaction time gap and model uncertainty to enhance the learning process, as demonstrated in our extensive experiments on multiple real-world datasets.

## Acknowledgments

This research was supported in part by an NSF IIS award IIS-1814450. The views and conclusions contained in this paper are those of the authors and should not be interpreted as representing any funding agency. We would like to thank the anonymous reviewers for their constructive comments.

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

# Appendix

**Organization of Appendix**    In this Appendix, we first summarize all notations used in our paper. Next we provide additional related work. Further, we provide the theoretical proof for Theorem 1, Lemma 1, and Lemma 2. Followed by that, we provide algorithms for the training and inference process. Further, we provide additional results for the extra baselines in terms of precision, NDCG@10 and NDCG@20. Finally, we provide a broader impact, limitation, future work, link to the source code, and checklist.

## A    Summary of Notations

We summarize the major notations used throughout the paper in Table 8.

Table 8:  Summary of Notations

| Notation | Description |
|---|---|
| $U, I$ | user set and item set |
| $u_t, i_t$ | user $u$ and item $i$ continuous representations at time $t$ |
| $\beta_{(u_t, i_t)}$ | total model uncertainty for user $u$ on item $i$ at time $t$ |
| $\alpha_{(u_t, i_t)}, \nu_{(u_t, i_t)}$ | model evidence for for user $u$ on item $i$ at time $t$ |
| $\gamma_{(u_t, i_t)}$ | predicted score for user $u$ on item $i$ at time $t$ |
| $\hat{r}_{(u_t, i_t)}, r_{(u_t, i_t)}$ | Uncertainty-aware predicted rating and ground truth for user $u$ on item $i$ at time $t$ |
| $\hat{r}_{(i_t, j_t)}$ | Predicted rating score difference between ground truth item $i$ and negative item $j$ for user $u$ at time $t$ |
| $\Theta$ | Overall model parameters i.e. $\Theta = (\psi, \omega, \theta, \phi)$ |
| $\psi, \omega, \theta, \phi$ | SDE drift, SDE diffusion, rating, and monotonic networks parameters |
| $\Delta t$ | interaction time gap between two consecutive items |
| $w_{(i_t, j_t)}$ | Weight coefficients for negative item $j$ based on uncertainty-aware predicted scores |
| $\tau$ | Threshold for uncertainty-aware predicted rating score |
| $\mathcal{N}_t$ | Negative items at time $t$ |
| $\mathcal{U}_{(u_t, i_t)}$ | Epistemic uncertainty for user $u$ and item $i$ interaction at time $t$ |
| $\lambda, \eta, \zeta$ | Balancing coefficient for EDL regularizer, uncertainty-aware rating, and WBPR loss respectively |

## B    Additional Related Work

In this section, we discuss some commonly used reocmmendation models to complement the related works covered in the main paper.

**Static and dynamic recommendation models.**    Matrix Factorization (MF) leverages user and item latent factors to infer user preferences [23, 13, 21]. MF is further extended with Bayesian Personalized Ranking (BPR) [30] and Factorization Machine (FM) [29]. Recently, deep learning-based recommender systems [8, 15] have achieved impressive performance. DeepFM [15] integrates traditional FM and deep learning to learn low- and high-order feature interactions. Both wide and deep networks are jointly trained in [8] for better memorization and generalization. The dynamic model shifts latent user preference over time to incorporate temporal information. TimeSVD++ [22] considers time-specific factors, which uses additive bias to model user and item-related temporal changes. Gaussian state-space models introduce time-evolving factors with a one-way Kalman filter [14]. To process implicit data, Sahoo et al. extended the hidden Markov model [31], and Charlin et al. [6] further augmented it with the Poisson emission. Nevertheless, these models grasp the evolving preferences of users, yet they exhibit limited foresight into future interactions, leading to poor recommendations performance.

**Graph-based recommendation models.**    Another popular line of recommendation systems is graph-based models. A graph captures high-order user-item interactions through an iterative process to provide effective recommendations [17]. Users and items are represented as a bipartite graph in [5] and links are predicted to provide recommendations. Similarly, a graph-based framework called Neural Graph Collaborative Filtering (NGCF) [39] explicitly encodes the collaborative signal in the form of high-order connectivities in a user-item bipartite graph via embedding propagation. However, these methods are unable to capture long-term user preferences or deal with cold-start problems.

# C Proof of Theoretical Results

In this section, we provide proofs of the theoretical results in the main paper.

## C.1 Proof of Theorem 1

*Proof.* Given Definition 1, we can show that the increase in interaction time gap $(t_2 - t_1)$ increases the chance that the user may deviate from the current interest. It turns out that the last term of Eq (6) has a larger variance, *i.e.,* there is a higher deviation in the user interest, if there is a longer time gap in the interaction. Suppose $t_3 > t_2$, then we have

$$\text{Var}\left[\int_{t_1}^{t_3} g(u(t), t; \omega) dB_t\right] = \mathbb{E}\left[\left(\int_{t_1}^{t_3} g(u(t), t; \omega) dB_t\right)^2\right] = \mathbb{E}\left[\int_{t_1}^{t_3} g(u(t), t; \omega)^2 dt\right]$$

$$= \mathbb{E}\left[\int_{t_1}^{t_2} g(u(t), t; \omega)^2 dt + \int_{t_2}^{t_3} g(u(t), t; \omega)^2 dt\right] \geq \mathbb{E}\left[\int_{t_1}^{t_2} g(u(t), t; \omega)^2 dt\right] \qquad (14)$$

$$= \text{Var}\left[\int_{t_1}^{t_2} g(u(t), t; \omega) dB_t\right]$$

where we applied the Ito Isometry in the second step. This result indicates that if there is a longer interaction gap $t_3 - t_1$, then the state is more uncertain. $\qquad \square$

## C.2 Proof of Lemma 1

Given the user final representation $u_t$, item final representation $i_t$, and interaction time gap $\Delta t$. The monotonic network produces $\beta_{(u_t, i_t)}$ as:

$$\beta_{(u_t, i_t)} = (W) \,\text{concat}(u_t, i_t, \Delta t)$$

$$= \begin{bmatrix} W_u^T & W_i^T & W_{\Delta t} \end{bmatrix} \begin{bmatrix} u_t \\ i_t \\ \Delta t \end{bmatrix}.$$

Hence $\frac{\partial \beta}{\partial(\Delta t)} = W_{\Delta t} \geq 0$.

## C.3 Proof of Lemma 2

Given the same setup as Lemma 2, the monotonic network outputs $\alpha_{(u_t, i_t)}$, and $\nu_{(u_t, i_t)}$. We first consider $\alpha_{(u_t, i_t)}$:

$$\alpha_{(u_t, i_t)} = [h_L \circ g \circ h_{L-1} \circ g \circ \cdots \circ g \circ h_2 \circ g \circ h_1](\text{concat}(u_t, i_t, \Delta t)),$$

where $h_i(x)$ is the transformation induced by the $i$th affine layer, $h_i(x) = W_i x + b_i$, $\circ$ is the composition of functions, and $g$ is an increasing activation function.

According to the backpropagation, we have

$$\nabla_{\text{concat}(u_t, i_t, \Delta t)} \alpha_{(u_t, i_t)}$$

$$= \begin{bmatrix} W_u^T \\ W_i^T \\ W_{\Delta t}^T \end{bmatrix} \left[ W_2^T \left[ W_3^T \left[ \cdots \left[ W_{L-1}^T \left[ W_L^T \odot g'(h_{L-1}) \right] \odot g'(h_{L-2}) \right] \cdots \right] \odot g'(h_2) \right] \odot g'(h_1) \right],$$

where $\odot$ is element-wise product of two matrices with the same size. The matrix $\begin{bmatrix} W_u^T \\ W_i^T \\ W_{\Delta t}^T \end{bmatrix}$ is just $W_1^T$,

where we rewrite $W_1^T$ the same blocks as that in Lemma 2. Therefore,

$$\frac{\partial \alpha_{(u_t, i_t)}}{\partial(\Delta t)} = W_{\Delta t}^T \left[ W_2^T \left[ W_3^T \left[ \cdots \left[ W_{L-1}^T \left[ W_L^T \odot g'(h_{L-1}) \right] \odot g'(h_{L-2}) \right] \cdots \right] \odot g'(h_2) \right] \odot g'(h_1) \right].$$

Since the entries in $W_1$, ..., $W_{L-1}$ are all negative, and $g'(h_1)$, ..., $g'(h_{L-1})$ are all positive. Therefore, if entries in $W_L$ are all negative, we have $\frac{\partial \alpha_{(u_t, i_t)}}{\partial(\Delta t)} \leq 0$.

Similarly, we can show that $\frac{\partial \nu}{\partial(\Delta t)} \leq 0$. We leverage the monotonic non-linear activation function i.e. $ELU(.)$ to satisfy this condition.

Table 9: Higher Order top-N metric comparisons with several baselines

| Models | MovieLens-1M | | | | Amazon Book | | | |
|---|---|---|---|---|---|---|---|---|
| | P@10 | P@20 | N@10 | N@20 | P@10 | P@20 | N@10 | N@20 |
| BERT4Rec[35] | 0.7258 | 0.8370 | 0.5131 | 0.5545 | 0.6583 | 0.7671 | 0.4833 | 0.5014 |
| ResAct[42] | 0.7443 | 0.8572 | 0.5276 | 0.5697 | 0.6759 | 0.7863 | 0.4976 | 0.5160 |
| GRU-ODE[16] | 0.7392 | 0.85156 | 0.5226 | 0.5645 | 0.6732 | 0.7830 | 0.4911 | 0.5095 |
| DiffuRec[26] | 0.7478 | 0.8616 | 0.5336 | 0.5759 | 0.6805 | 0.7868 | 0.4994 | 0.5179 |
| TiSASRec[24] | 0.7401 | 0.8532 | 0.5297 | 0.5717 | 0.6813 | 0.7889 | 0.4951 | 0.5135 |
| STOSA[12] | 0.7484 | 0.8622 | 0.5310 | 0.5733 | 0.6788 | 0.7902 | 0.5025 | 0.5210 |
| DuoRec[28] | 0.7489 | 0.8659 | 0.5259 | 0.5682 | 0.6831 | 0.7945 | 0.5001 | 0.5186 |
| GDERec[27] | 0.7503 | 0.8649 | 0.5324 | 0.5750 | 0.6808 | 0.7928 | 0.5077 | 0.5264 |
| **E-NSDE** | **0.7745** | **0.8926** | **0.5467** | **0.5907** | **0.7021** | **0.8175** | **0.5198** | **0.5390** |

# D   Training and Inference Processes

The training procedure involves the end-to-end parameter updates associated with the NSDE module and evidential module. Both modules utilize the overall loss mentioned in Eq (13), which includes a supervised signal from evidential loss and ranking loss from WBPR loss. The rating network ($\theta$) and monotonic network ($\phi$) are updated with the Adam optimizer, and the SDE module with model parameters ($\psi, \omega$) is updated with SDE adjoint method [25]. Algorithm 1 shows the training process that learns the model parameters. During inference, we consider test users (*i.e.,* distinct users from the training set) and perform standard sequential recommendations respecting the time interval of interactions.

---

**Algorithm 1** E-NSDE Training

---

**Require:** Hyperparameters: $\lambda, \eta, \zeta, \alpha, \beta, \gamma, \nu$
  Initialize both NSDE and EDL modules:$\Theta = (\psi, \omega, \theta, \phi)$
  **while** not converge **do**
    Sample train user $\mathcal{T}_u$ from user pool $U$
    **for** all $u \in \mathcal{T}_u$ **do**
      Compute user and item final representations using Eq (6) and Eq (7) respectively from SDE module.
      Compute interaction time gap $\Delta t$ with target item
      Compute EDL loss for each sequence using Eq (10).
      Compute weighted BPR loss for each sequence using Eq (12).
      Perform end-to-end update using overall loss Eq (13)
    **end for**
  **end while**

---

# E   Additional Experimental Results

**Higher order top-$N$ results.** We report the results using top-$N$ metric with a larger $N$ value (i.e., @10 and @20) in Table 9. We reported those results on two sparse datasets: Movielens-1M with 95.75% sparsity and Amazon Book with 99.98% sparsity. In both datasets, our proposal models outperform other baselines in those metrics and show consistent performance over metrics.

**Additional datasets.** We conduct experiments on two additional datasets from the book and music domains to cover a broader range. The Book-Crossing dataset is a collection of book ratings, including both explicit ratings (1–10 stars) and implicit ratings (interaction) [46]. The Yahoo! Music dataset is a collection of user ratings for songs, albums, artists, and genres, with fine-resolution timestamps for ratings [10]. The proposed E-NSDE consistently outperforms the competitive baselines.

Table 10: Recommendation Performance on More Datasets

| Model | BookCrossing | | Yahoo Music | |
|---|---|---|---|---|
| | P@5 | nDCG@5 | P@5 | nDCG@5 |
| BERT4Rec | 0.4811 | 0.4585 | 0.5269 | 0.4773 |
| GRU-ODE | 0.4965 | 0.4662 | 0.5316 | 0.4825 |
| **E-NSDE** | **0.5217** | **0.4834** | **0.5621** | **0.5145** |

## F  Broader Societal Impact, Limitation, and Future Work

This paper presents a novel stochastic neural ODE and evidential learning-based recommendation model to capture users' continuously evolving behavior. This work aims to understand the user's dynamic interest considering an actual time interval of user interaction with the system. This is very useful in many time-series user interaction fields like e-commerce, social media, health care, and gaming sectors.

As a potential limitation of this work, if user-item interactions have uniform time intervals like in click-through, then the impact of time-aware interval is less effective in capturing uncertainty.

Future work could be applying the E-NSDE method to other areas of recommender systems, such as healthcare, where patient behaviors evolve over time, and capturing their real-time behavior is crucial to handling their problem. Similarly, understanding user behavior in the gaming sector is also a potential area to explore in the future.

## G  Source Code

For the source code, please click this link: `https://github.com/ritmininglab/ENSDE`

