# OpenReview forum: "Evidential Stochastic Differential Equations for Time-Aware Sequential Recommendation"
_NeurIPS.cc/2024/Conference — NeurIPS 2024 poster_

### Official Review · Reviewer_zE4M · 2024-07-09

**Soundness:** 2
**Presentation:** 3
**Contribution:** 2
**Rating:** 4
**Confidence:** 4

**Summary:**

The authors note that existing methods assume a uniform time interval among user behaviors. This paper posits that the time intervals in sequential recommendation increase the uncertainty of users' behavior. Therefore, it proposes NSDE to learn users’ fine-grained time-evolving behavior, while evidential learning quantifies both aleatoric and epistemic uncertainties. The authors believe that this is something other sequential learning methods have not considered.

**Strengths:**

1. The paper starts from the uncertainty of time intervals to capture the evolution and uncertainty of user behavior, which is a valuable insight. However, this perspective lacks innovation, as "Sequential recommendation via stochastic self-attention" (2022) also approaches the problem from the same insight.
2. The straightforward derivation of the relationship between increasing time intervals and uncertainty to guide learning demonstrates a degree of innovation and robustness in modeling.

**Weaknesses:**

1.	There are formatting issues in lines 99-105. The abbreviation for Normalized Discounted Cumulative Gain is generally NDCG.The combination of metrics chosen in the paper is uncommon and lacks persuasiveness. According to line 294, the metrics are derived from Bert4Rec. However, Bert4Rec's metrics are Hit Ratio (HR), Normalized Discounted Cumulative Gain (NDCG), and Mean Reciprocal Rank (MRR).

2.The selected baselines could include more recent state-of-the-art models, such as STOSA, DuoRec, CoSeRec, GDERec,TiCoSeRec and GCG-ODE. The baselines particularly lack ODE-related sequential recommendation models. Among the two ODE-related models, LT-OCF is not specifically designed to address sequential recommendation.

3.The increase in time intervals may lead to changes in user interests. Addressing this issue within NODE-based recommender systems, the paper proposes corresponding solutions and offers a relatively novel perspective.

**Questions:**

1.Why do the authors believe that sequential recommendation methods typically use uniform time intervals and overlook changes in user interests? Many earlier works have addressed this issue, such as "Uniform Sequence Better: Time Interval Aware Data Augmentation for Sequential Recommendation".
2.See weeknesses

---

> ### Author Rebuttal · Authors · 2024-08-07
>
> **Q1: The abbreviation for Normalized Discounted Cumulative Gain is generally NDCG.The combination of metrics chosen in the paper is uncommon and lacks persuasiveness. According to line 294, the metrics are derived from Bert4Rec. However, Bert4Rec's metrics are Hit Ratio (HR), Normalized Discounted Cumulative Gain (NDCG), and Mean Reciprocal Rank (MRR).**
>
> Thanks for the suggestion to clarify the metrics. We follow a similar setup in Bert4Rec as a next-item prediction. However, we selected three popular metrics (i.e., Precision, Recall, and NDCG) that are widely used in evaluating recommendation models. Here, Precision is equivalent to Hit Ratio (HR), and due to next-item prediction, it is also equivalent to Recall. Further, NDCG is a metric that considers position by assigning greater weights to higher positions. We will clarify these in the revised paper.
>
> **Q2: The selected baselines could include more recent state-of-the-art models, such as STOSA, DuoRec, CoSeRec, GDERec, TiCoSeRec and GCG-ODE. The baselines particularly lack ODE-related sequential recommendation models. Among the two ODE-related models, LT-OCF is not specifically designed to address sequential recommendation.**
>
> Thanks for pointing out recent baselines. Limited by time, we have provided results for three of the additional baselines, including STOSA, DuoRec, and GDERec, and compared them with our approach (please refer to the table presented in the overall rebuttal to all reviewers). The results clearly show that the proposed E-NSDE model outperforms these baselines by effectively leveraging the interaction time gap in novel ways to offer uncertainty-aware recommendations with diverse items that better align with the user interest. We will provide a more complete comparison in the revised paper.
>
> **Q3: Why do the authors believe that sequential recommendation methods typically use uniform time intervals and overlook changes in user interests? Many earlier works have addressed this issue, such as "Uniform Sequence Better: Time Interval Aware Data Augmentation for Sequential Recommendation".**
>
> Thank you for the comment. In the paper suggested by the reviewer, the authors propose to address the issue of varying time intervals for sequential recommendations. Instead of leveraging the varied intervals, it argues that sequences with uniformly distributed time intervals are more beneficial for performance improvement and introduces new approaches to transform non-uniform sequences into uniform ones. While there may also be other works that consider time intervals in user interactions, none of these works address the increasing uncertainty arising from the extended interaction intervals. We believe our paper is the first work that establishes a correlation between the interaction time interval and the model uncertainty, and leverages this important connection to improve recommendations through time-aware uncertainty guided exploration.

---

> > ### Author Response · Authors · 2024-08-12
> >
> > Dear Reviewer zE4M,
> >
> > Thank you once again for your insightful comments and questions. In our rebuttal, we have:
> >
> > - provided a comparison with the suggested baselines (STOSA, DuoRec, and GDERec).
> >
> > - compared the paper "Uniform Sequence Better" with our work, focusing on the varying uniform time intervals of user interactions.
> >
> > - offered a detailed explanation of the metrics used.
> >
> > We believe that addressing your comments will improve the paper's readability, and we appreciate your support. We hope you find our responses satisfactory and consider updating the score accordingly. We are more than happy to address any further questions you may have.

---

### Official Review · Reviewer_V8wq · 2024-07-12

**Soundness:** 2
**Presentation:** 3
**Contribution:** 2
**Rating:** 4
**Confidence:** 3

**Summary:**

This paper investigates the modeling of interaction time intervals in sequential recommendation. Considering both time interval and model uncertainty, this paper formulates E-NSDE to integrate NSDE and evidential learning to model effective time-aware sequential recommendation.  Experimental results on four real-world datasets prove the effectiveness of proposed method.

**Strengths:**

1. The problem is interesting. The time interval between interactions is really an important factor in the sequential recommendation. Meanwhile, time-aware uncertainty guides the recommendation to adapt to users' changing preferences.

2. Combining two existing techniques is well-motivated. Overall, both user NSDE and item NSDE are introduced here to encode the evaluation of the preference and inherent noise. The evidential module then provides uncertainty-aware rating prediction. All the stack techniques are combined reasonably.

3. Experiments are conducted to verify the effectiveness. And the results are statically reliable.

**Weaknesses:**

1. The motivation in the introduction is unclear. As the authors argue, GRU-ODE recommends genres that come from past behaviors, while NSDE recommends genres which have potential future benefits in Table 2. It is not sure if some genres recommended by NSDE are in accordance with long-term interests. For example, Sci_Fi, Mystery, and Crime in Table 2. My concern is whether motivation is not the real need of the user.

2.  As to method, in Figure 1, users also have evolved into representation. It is clear that the user interfaces with items in an evolving manner. But not so clear that the users also evolved into different user.

3. The experiments in Table 4 report the P@5, nDCG@5. However, in the movielens datasets, 1m and 100k are used, so more top-n results should be given to verify the effectiveness. In line 316, "Uncertainty vs. Interaction gap vs." may be a typo.

**Questions:**

1. Please refer to weakness 1, and give more clarification on Table 2.

2. Please explain the relationship between this work and [1],[2],[3]. The baselines lack the time-interval awarded by the sequential model and diversity model.

3. Pleas refer to weakness3, more n in top-n results should be given.



[1]Time interval aware self-attention for sequential recommendation,WSDM20

[2]Learning Graph ODE for Continuous-Time Sequential Recommendation,TKDE

[3]Temporal Conformity-aware Hawkes Graph Network for Recommendations,Webconf 24

**Limitations:**

Yes

---

> ### Author Rebuttal · Authors · 2024-08-07
>
> **Q1: The motivation in the introduction is unclear. As the authors argue, GRU-ODE recommends genres that come from past behaviors, while NSDE recommends genres which have potential future benefits in Table 2. It is not sure if some genres recommended by NSDE are in accordance with long-term interests. For example, Sci\_Fi, Mystery, and Crime in Table 2. My concern is whether motivation is not the real need of the user.**
>
> Thank you for the comment. We would like to clarify that we included motivating examples (in Table 2 of the main paper) to demonstrate the need for an effective means of exploration, which is essential and critical to uncover the user's long-term interests. Especially, if a user is inactive for a longer period, maintaining their engagement in the system requires more aggressive exploration, as indicated by the larger time-interval gap in Table 2. While there is no total guarantee that every explored item will be successful, our epistemic uncertainty-guided exploration (see Eq. 11) allows the model to recommend items that are largely unknown to the users (due to the second term) while potentially interesting to them (due to the first term). Both our quantitative (shown in Table 4) and qualitative results (shown in Table 5a) clearly demonstrate the effectiveness of our approach.
>
>
> **Q2: As to method, in Figure 1, users also have evolved into representation. It is clear that the user interfaces with items in an evolving manner. But not so clear that the users also evolved into different user.**
>
> We would like to clarify that our NSDE module consists of two key components: diffusion and drift. The diffusion component captures the user's extrinsic behavior, while the drift component captures the user's intrinsic behavior. This module is responsible for modeling the user's evolving interests over time. Additionally, as the reviewer pointed out, the user-item interface also evolves over time, and the uncertainty of these interactions is captured through the evidential module.
>
> **Q3: More top-n results**
>
> Thanks for the suggestion. We have reported additional higher-order top-N (@10 and @20) results in the table presented in the general response to all reviewers.
>
>
> **Q4: Please explain the relationship between this work and [1],[2],[3]. The baselines lack the time-interval awarded by the sequential model and diversity model.**
>
> Thanks for pointing out those papers.
>
> For prior efforts on time interval-aware recommendations, [1] introduces a self-attention mechanism that considers both the absolute positions of items and the time intervals between them in interaction history, in order to allow the system to be aware of the influence of different time intervals on predicting the next item. However, the architecture leverages the black-box embedding and attention mechanisms for time intervals. In comparison, the proposed approach introduces differential equations and monotonic networks to explicitly encourage the system to generate higher uncertainty to longer time intervals.
>
> [2] is published in July 2024, and therefore, it is impossible for us to be aware of it during submission. [2] introduces a Graph Ordinary Differential Equation model to capture continuous-time dynamics in sequential recommendation systems. The key differences between [2] and the proposed method are: 1) Ordinary differential equations are deterministic, while our approach leverages stochastic differential equations, which involve a noisy process to model how the uncertainty of users' and items' representation evolves over time.
> 2) The proposed approach explicitly encourages the system to assign higher uncertainty to longer time intervals.
>
> [3] introduces Hawkes processes to model the intensity of user interactions and leverages graph neural networks to learn from the complex user-item interactions. Hawkes process is a point process to model the occurrence of sequential events with irregular inter-arrival time. Unlike the proposed method, it does not explicitly model the uncertainty or enforce higher uncertainty to longer time intervals.
>
> We will incorporate them into the related work section of the revised paper. Further, we have provided results for TiSASRec ([1] Time interval aware self-attention for sequential recommendation, WSDM20) and compared with the proposed method in the table of the general response, which shows the proposed approach achieves a better recommendation performance.

---

> > ### Author Response · Authors · 2024-08-12
> >
> > Dear Reviewer V8wq,
> >
> > Thank you again for your insightful comments and questions. In our rebuttal, we have:
> >
> > - clarified the motivation behind the paper and emphasized the importance of exploring new tastes for users who have not been active for a long time.
> >
> > - explained the evolving nature of user behavior and how the drift and diffusion components of the NSDE module help capture these changes.
> >
> > - incorporated more top-n results on both datasets with competitive baselines.
> >
> > - explained relationships between suggested papers and further included one paper i.e. TiSASRec as a baseline comparison.
> >
> > We believe that addressing your comments will enhance the paper's readability, and we appreciate your support. We hope you find our responses satisfactory and consider updating the score accordingly. We are more than happy to address any further questions you may have.

---

> > > ### Comment · Reviewer_V8wq · 2024-08-13
> > >
> > > Thank you for your response; I will keep my score due to concerns about related work.

---

> ### Author Response · Authors · 2024-08-13
>
> Dear Reviewer V8wq,
>
> Thank you again for reading our responses and providing quick feedback. We would like to clarify that we have clearly described the fundamental differences between the three mentioned related works and ours. We have also experimentally compared the most relevant method i.e. TiSASRec, with our proposed ENSDE model and the result is shown in the table of the general response. We will appreciate if you could be more explicit about your remaining concerns about the related work.

---

### Official Review · Reviewer_LeEA · 2024-07-13

**Soundness:** 3
**Presentation:** 4
**Contribution:** 3
**Rating:** 5
**Confidence:** 4

**Summary:**

The paper revolves around enhancing sequential recommendation systems by incorporating time-awareness through the utilization of Evidential Neural Stochastic Differential Equations (E-NSDE). Traditional recommendation systems often overlook the temporal dynamics of user interactions, leading to suboptimal recommendations. The authors aim to bridge this gap by developing a novel framework that captures the evolving behavior of users over time intervals, thereby improving the accuracy and reliability of recommendations.
It acknowledges the limitations of existing methods in effectively modeling temporal dynamics and uncertainty in user behavior. By integrating NSDE and evidential learning, the authors build upon the foundations laid by previous works on stochastic differential equations and recommendation systems. This integration allows for a more robust modeling of user preferences and interactions over time, setting the stage for more accurate predictions.

**Strengths:**

1.	The E-NSDE framework dynamically models user behaviors over time, effectively addressing a crucial gap in traditional recommendation systems that often overlook temporal dynamics.
2.	By incorporating NSDEs to capture the continuous-time dynamics of user interactions, along with evidential learning to assess recommendation uncertainty, this dual approach ensures that the recommendations are not only time-sensitive but also carry a higher degree of confidence.
3.	The integration of NSDE and evidential learning into a unified framework facilitates comprehensive modeling of both the evolution of user preferences and the inherent uncertainty in predicting user behavior.

**Weaknesses:**

1.	When user-item interactions have uniform time intervals, such as in click-through scenarios, the model's time-aware approach may be less effective in capturing uncertainty.
2.	The paper acknowledges the interpretability of the E-NSDE framework but falls short in providing detailed explanations of the decision-making process and incorporating feature importance analysis, which would significantly enhance transparency and build user trust.
3.	The discussion regarding the generalizability of the E-NSDE framework across various domains and datasets is inadequate. Providing more detailed insights into the model's performance in diverse settings would clarify its robustness and broader applicability.

**Questions:**

1.	Could the authors provide a detailed explanation of how NSDE and evidential learning are integrated within the E-NSDE framework?

2.	Can the authors elaborate on the characteristics of the real-world datasets used in their experiments? How diverse are these datasets, and do they encompass a broad spectrum of recommendation scenarios to validate the generalizability of the E-NSDE framework?

3.	Are there specific techniques or methodologies incorporated to enhance the transparency and explainability of the recommendation process?

4.	In the context of large-scale recommendation systems, how does the E-NSDE framework address challenges related to scalability and computational efficiency? Have any experiments been conducted to evaluate the framework's scalability as dataset sizes or user interactions increase?

**Limitations:**

The authors have adequately addressed the limitations and identified no potential negative societal impacts of their work.

---

> ### Author Rebuttal · Authors · 2024-08-07
>
> **Q1: When user-item interactions have uniform time intervals, such as in click-through scenarios, the model's time-aware approach may be less effective in capturing uncertainty.**
>
> We agree with the reviewer that when interaction intervals are uniform, the time-aware uncertainty component is less effective. We have pointed out this as we discuss the limitation of our approach in Appendix G. Meanwhile,  we would like to clarify that the proposed model's performance will not degrade when interaction intervals are uniform when compared to existing state-of-the-art models.
>
> **Q2: The paper acknowledges the interpretability of the E-NSDE framework but falls short in providing detailed explanations of the decision-making process and incorporating feature importance analysis, which would significantly enhance transparency and build user trust.**
>
> Thank you for the suggestion; it would indeed be valuable to examine details at the feature level. However, we would like to emphasize that our approach aims to improve interpretability at the module level using epistemic uncertainty. In particular, our epistemic uncertainty indicates the model's confidence in recommending the next item, which increases the transparency of the recommendation process and can help build user trust.
>
> **Q3: Providing more detailed insights into the model's performance in diverse settings would clarify its robustness and broader applicability.**
>
> Thank you for the great suggestion! The datasets used in our experiments indeed cover very diverse settings. First, the datasets possess different levels of sparsity that varies from 95\% to over 99.9\%: Movielens-1M with a sparsity of 95.75\%, Netflix with a sparsity of 98.82\%, and Amazon Books with a sparsity of 99.98\%. The level of sparsity is usually tied to the difficulty of the dataset when making recommendations. Meanwhile, different datasets also encode diverse interaction behaviors from users. For example, the interaction patterns vary significantly based on the timing of interactions. In the Movielens-1M dataset, most interactions occur within seconds, minutes, and days. For Netflix, interactions predominantly happen with day-long gaps. However, in the Amazon Books dataset, user interactions span months or even years. We selected these datasets to particularly demonstrate the proposed model's effectiveness across different interaction time intervals. Our model exhibits robust performance across all these datasets, providing evidence of its broad applicability in any time-aware user-interaction settings.
>
>
> **Q4:Detailed explanation of how NSDE and evidential learning are integrated within the E-NSDE framework**
>
> Our NSDE module generates richer user and item representations by explicitly considering their interaction time. Those representations are fed to the EDL module to capture interaction uncertainty and generate predicted scores, as shown in Figure 1 of the main paper. We further leverage a monotonic network in building the relationship between the interaction time gap and model uncertainty. The monotonic network ensures the increase of the uncertainty (i.e., variance) of the predicted rating along with the increase of time interval ($\Delta t$). This novel integration leads to an end-to-end E-NSDE framework that provides an effective time-aware sequential recommendation model.
>
> **Q5: Elaborate on the characteristics of the real-world datasets used in their experiments? How diverse are these datasets, and do they encompass a broad spectrum of recommendation scenarios to validate the generalizability of the E-NSDE framework?**
>
> Please refer to the answer to Q3.
>
> **Q6: Are there specific techniques or methodologies incorporated to enhance the transparency and explainability of the recommendation process?**
>
> A key component to enhance transparency and explainability of the recommendation process is the evidential learning module, which allows us to perform fine-grained uncertainty decomposition. As a result, the decomposed epistemic uncertainty indicates the model's confidence in recommending the next item, which increases the transparency of the recommendation process and can help to build user trust.
>
>
> **Q7: In the context of large-scale recommendation systems, how does the E-NSDE framework address challenges related to scalability and computational efficiency? Have any experiments been conducted to evaluate the framework's scalability as dataset sizes or user interactions increase?**
>
> We would like to clarify that our training setup leverages incremental time computation for the stochastic differential equation, progressing from the previous time step $t-1$ to the current step $t$ rather than starting from the beginning ($t_0$). This approach, which uses the previous step's computation for the next step, enhances both the forward and backward passes of the NSDE module. Consequently, this improves the model's scalability, as demonstrated by its applicability to datasets ranging from smaller ones(e.g., Movielens) to larger ones (e.g., Netflix and Amazon). For more details on the training setup, please refer to the experimental setting section in the main paper.

---

> > ### Author Response · Authors · 2024-08-12
> >
> > Dear Reviewer LeEA,
> >
> > Thank you once again for your thoughtful comments and questions. In our rebuttal, we have:
> >
> > - explained the impact of capturing uncertainty when user interactions occur at uniform time intervals,
> >
> > - provided a detailed explanation of the interpretability, generalizability, and transparency aspects of our work, specifically focusing on the two key components of the ENSDE model.
> >
> > - offered a thorough discussion on handling sparse datasets and the diverse settings of our experiments.
> >
> > We believe that addressing your suggestions has significantly strengthened our paper, and we appreciate your support. We hope you find our responses satisfactory and consider updating the score accordingly. We are more than happy to answer any additional questions you may have.

---

### Official Review · Reviewer_p7WH · 2024-07-13

**Soundness:** 3
**Presentation:** 3
**Contribution:** 3
**Rating:** 7
**Confidence:** 2

**Summary:**

This work investigates sequential recommendation, and proposes a new method that utilizes stochastic differential equations (SDEs) to model dynamic time intervals and estimate uncertainty.

Overall, this study addresses an engaging problem and provides a novel and reasonable solution. Extensive experiments have been conducted. Consequently, I lean towards accept.

**Strengths:**

1.	This work studies on an interesting and important problem --- how to capture dynamic time interval as well as estimate model uncertainty.

2.	The paper is well-written, with clear motivations.

3.	The application of stochastic differential equations to model sequential recommendation is both novel and reasonable.

4.	Extensive experiments are conducted to validate the effectiveness of the proposal.

**Weaknesses:**

1.	It would be advantageous to include diffusion model-based sequential recommendation baselines for comparison in the experiments, such as [a1][a2], especially since SDEs have been utilized in these methods as well.

2.	The work reports performance metrics only for P@5 and NDCG@5. It would be better to include the results with different @N, particularly @20, which is commonly adopted by recent work.

3.	A discussion on the limitations and future directions of the research in Section 6 would be beneficial.

4.	There are some typos. For example, in the eq.(13), ‘BPR’->’WBPR’; In the line 316, removing ‘vs’.


5.	Some important relate work is omitted:
[a1] TOIS’23: Diffurec: A diffusion model for sequential recommendation
[a2] NIPS’23: Generate what you prefer: Reshaping sequential recommendation via guided diffusion

**Questions:**

Please refer to weaknesses.

**Limitations:**

No concern.

---

> ### Author Rebuttal · Authors · 2024-08-07
>
> **Q1:  Diffusion model-based sequential recommendation baselines for comparison.**
>
> Thanks for suggesting those important related works. We will cite them and add a discussion in the revised paper. More specifically, DiffuRec [a1]  models item representations as distributions by corrupting the target item embedding into a Gaussian distribution through adding noise and reversing the Gaussian noise into the target item representation based on historical interactions. DreamRec [a2] formulates sequential recommendation as a learning-to-generate task. It uses a guided diffusion model with a transformer encoder to generate the distribution of representations from historical items and adds noise to items to explore the item space distribution.
> In summary, the two papers leverage explicit augmentation of noises to representations, which are technically different from the proposed method.
>
> As suggested by the reviewer, we have conducted experiments to compare with the diffusion model-based sequential recommendation, i.e., DiffuRec and the results are presented in the Table in the general response to all reviewers. Since DiffuRec represents compact item representation with construction and injection of uncertainty, it does not properly address continuous time-evolving aspects of user interest and hence has lower performance than the proposed E-NSDE model. Limited by time, we will include a comparison with DreamRec [a2] in the revised paper.
>
> **Q2:  Better to include the results with different @N, particularly @20**
>
> Thanks for the suggestion. We have conducted additional experiments and reported the top @N =10 and 20 results in the Table of the general response to all reviewers.
>
> **Q3:  Limitations and future directions of the research**
>
> Thanks for the suggestion. We have included a discussion of limitations and future work in Appendix G.

---

> > ### Author Response · Authors · 2024-08-12
> >
> > Dear Reviewer p7WH,
> >
> > Thank you once again for your insightful comments and questions. In our rebuttal, we've clarified the key differences of suggested baselines (DiffuRec and DreamRec) from our work and further included DiffuRec in baseline comparison with higher ranking metrics. We believe that addressing your feedback has significantly improved and strengthened our paper. We hope you find our responses satisfactory and are more than happy to address any further questions you may have.

---

### Author Rebuttal · Authors · 2024-08-07

We thank all the reviewers for their constructive comments and suggestions. Here, we provide results for several suggested baselines and top-$N$ metrics with a higher $N$ value as requested by the reviewers:

| **Datasets** | **Metric**   | **Bert4Rec** | **ResAct** |**GRU-ODE** |**DiffuRec** |**TiSASRec** |**STOSA** |**DuoRec** |**GDERec** |**E-NSDE** |
|--|--|--|--|--|--|--|--|--|--|--|
| *MovieLens-1M*     | P@5   |  0.4163   | 0.4286   | 0.4275 |  0.4212 |   0.4253 | 0.4314 | 0.4337    | 0.4301   | **0.4551** |
|    | P@10   | 0.7258    | 0.7443   |  0.7392|  0.7478 |0.7401    |0.7484 | 0.7489    |  0.7503  | **0.7745**|
|   | P@20  | 0.8370    | 0.8572   | 0.8515 | 0.8616  | 0.8532|  0.8622   |0.8659    |0.8649 |**0.8926**|
|    | NDCG@5   | 0.3754     |   0.3814  | 0.3792 |  0.3795 |0.3829    |0.3801 |   0.3852  |  0.3805  | **0.3982**|
|     | NDCG@10   | 0.5131    |  0.5276  |0.5226  |  0.5336  |0.5297    | 0.5310|   0.5259  |  0.5324  | **0.5467**|
|    | NDCG@20   |0.5545    | 0.5697  | 0.5645 | 0.5759    | 0.5717   | 0.5733|   0.5682  |  0.5750  | **0.5907**|
| *Amazon Book*      | P@5  |  0.3846    |0.3884    | 0.3856| 0.3870    | 0.3940 |0.3899 |   0.3902 |  0.3935  |**0.4021** |
|     | P@10  | 0.6583    |   0.6759 | 0.6732 |   0.6805  |  0.6813  | 0.6788|    0.6831 |    0.6808 |**0.7021** |
|       | P@20  | 0.7671    | 0.7863   | 0.7830  |   0.7868 | 0.7889| 0.7902    | 0.7945   |0.7928 |**0.8175** |
|      | NDCG@5  |  0.3463   | 0.3472| 0.3455 | 0.3415  | 0.3521  |0.3538|  0.3504  | 0.3563   | **0.3621**|
|      | NDCG@10  |  0.4833   | 0.4976   | 0.4911 |    0.4994    |0.4951 |   0.5025  |    0.5001| 0.5077| **0.5198**|
|      | NDCG@20  | 0.5014    | 0.5160   |0.5095  |   0.5179  | 0.5135   | 0.5210| 0.5186    |  0.5264  |**0.5390** |

---

### Decision · Program_Chairs · 2024-09-25

**Decision:**

Accept (poster)

**Comment:**

Very slightly on the positive side according to review scores, though arguably mostly saved by one reviewer. Reviewers find the problem interesting, straightforward, and reasonably convincing in terms of experiments. Reviewers complain about interpretability issues, lack of metrics (and other experimental details), and some problems with the motivation.

All of the reviews (including the most positive one) aren't particularly detailed, so it's hard to be confident of the recommendation. Not that that's the authors' fault.